# Vision Transformers Secretly Crave Noise

## Abstract

Data augmentation and regularization have proven to be fundamental techniques for enhancing the generalization of deep neural networks. While canonical methods such as RandAug, CutMix, Mixup, RandErase, and DropPath offer diverse regularization effects, their combined use appears to have reached a saturation point, leaving little room for further performance gains. In this work, we introduce `DiffNoise`, a novel data augmentation strategy that injects smooth noise-based perturbations into the input embedding space rather than directly into the raw input. Contrary to the conventional belief, `DiffNoise` performs orthogonally to existing data augmentations, improving the standard recipe that has largely reached saturation. This improvement may be interpreted as expanding the augmentation space along a previously unexplored axis, without any architectural modifications or auxiliary objectives. Furthermore, `DiffNoise` implicitly benefits from a more improved localization capability and learn generalized, robust representations across various models. Extensive experiments across a wide spectrum of model families—including ViTs, CLIP, and self-supervised architectures—show that `DiffNoise` consistently enhances performance across multiple downstream tasks. Code is available in the Supplementary Material.

## 1 Introduction

Data augmentation is a staple for the generalization capability of vision models. It has delivered large gains in CNNs (He et al., 2016) by enriching data diversity and mitigating overfitting. After the emergence of large-scale Vision Transformers (ViT) (Dosovitskiy et al., 2020) and Transformer-based training (He et al., 2022; Xie et al., 2022; Wei et al., 2023; Zheng et al., 2023; Choi et al., 2024b), they have made augmentation even more crucial. This is presumably due to ViTs having a higher capacity, which relies on global self-attention, lacking spatial inductive biases, and making them sensitive to small perturbations. In downstream fine-tuning with ViTs, this issue becomes central: effective transfer of pre-trained parameters depends on carefully chosen augmentations that inject task-relevant inductive biases and curb overfitting under limited labels.

Tremendous efforts have been made to discover a golden augmentation recipe, but we now face *the wall of the standard setup* using CutMix (Yun et al., 2019), Mixup (Zhang et al., 2017), Drop-Path (Huang et al., 2016), and RandAug (Cubuk et al., 2020) for fine-tuning ViTs (often combined with Label Smoothing and Dropout). In practice, despite broad adoption and strong results, its gains have reached a performance plateau[1]. Fig. 1 (a) illustrates that stacking additional augmentations on $\mathcal{R}_b$ yields negligible improvement. Why does the standard recipe saturate? Our insight is that most existing data augmentations largely overlap in the augmentation space, which *limits the complementarity between them*. At times, we argue fine-tuned ViTs seem weaker in fine-detail localization and robustness, likely due to this.

To move beyond saturated axes, an ingenious path may lie in generative training, where corruption by noise plays a core role. Language models (Hua et al., 2021; Nukrai et al., 2022; Jain et al., 2024) and diffusion-based methods (Ho et al., 2020; Song et al., 2020; Nichol & Dhariwal, 2021; Rombach et al., 2022; Ramesh et al., 2021; Saharia et al., 2022) inject noise at embedding-level for improved feature learning in pretraining and downstream tasks (Ho et al., 2020; Wang et al., 2022; Choi et al.,

---

[1]Prior works (Tan & Le, 2019; Touvron et al., 2021; Han et al., 2021; Wightman et al., 2021; Steiner et al., 2021; Touvron et al., 2022; Dehghani et al., 2023; Kim et al., 2024; Heo et al., 2025) employed or slightly extended the standard setup, yet the combination remains the default and has plateaued in augmentation diversity.

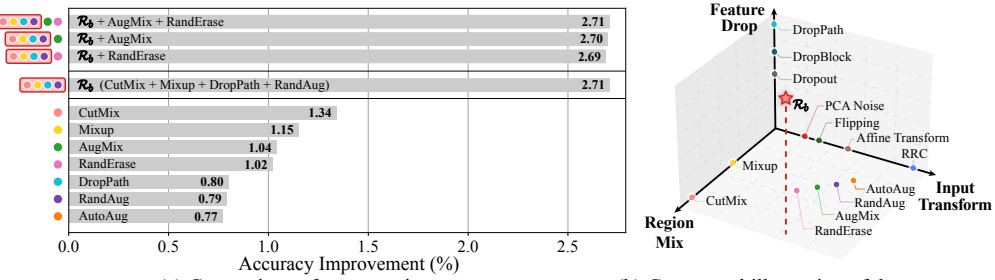

(a) Comparison of accuracy improvements

(b) Conceptual illustration of the aug. space

Figure 1: **Where we stand? Data augmentation may reach its limits.** When more is no more, we are now at *the wall of augmentations* that limits further progress. Current ImageNet training appears to have reached a point where the de facto setup dominates all alternatives. Specifically, **(a)** ImageNet-1K accuracy gains vanish beyond the de facto recipe $\mathcal{R}_b$ = CutMix, Mixup, DropPath, and RandAug (often combined with Label Smoothing and Dropout). **(b)** Conceptually, augmentation space can be drawn in three explicit axes—*input transforms*, *region-level mixing*, *feature-level dropping*. We argue that methods on these axes overlap in regularization, which could visually explain the saturation in (a).

2024a). Drawing from this insight, we introduce `DiffNoise`[2], a simple embedding-level noise augmentation method for improved representation learning. We argue `DiffNoise` complements the standard setup $\mathcal{R}_b$ in an orthogonal manner and integrates effectively as a plug-in. What is more, it turns out that (1) `DiffNoise` enhances localization by suppressing high-norm values, leading to broader attention with improved fine detail sensitivity; (2) `DiffNoise` acts like a embedding- (*i.e.,* token-) level regularization due to injecting noise in the embedding space, thus leading to improved robustness; (3) `DiffNoise` demonstrates the ability to learn generalized and robust representations.

We conduct extensive experiments across a diverse suite of architectures with `DiffNoise`. Our evaluation spans ViT-{S,B,L} (Dosovitskiy et al., 2020), multi-modal pre-trained models (CLIP (Radford et al., 2021)), and modern self-supervised learning (SSL) frameworks (Masked Image Modeling (He et al., 2022; Xie et al., 2022) and Diffusion-based Masked Image Modeling (Wei et al., 2023; Zheng et al., 2023; Choi et al., 2024b)). When combined with existing augmentations, `DiffNoise` achieves up to 4.41%p improvement in performance across various settings. We validate beyond ViTs—testing `DiffNoise` on CNNs (ResNet-26/50) and hierarchical transformer (Swin V2-L)—and observe consistent gains, indicating that the mechanism is not tied to a particular backbone design.

We further report the evaluations on downstream tasks, including image classification, fine-grained visual classification (FGVC), semantic segmentation, object detection, and instance segmentation. Notably, tasks that hinge on subtle, part-level cues—*e.g.,* FGVC—benefit the most: as a strong localizer, `DiffNoise` sharpens sensitivity to fine detail by suppressing attention sinks and shortening effective attention distance, translating into consistent gains on fine-grained benchmarks (Wah et al., 2011; Van Horn et al., 2015).

## 2 METHOD

### 2.1 BACKGROUND

**Embedding-level noise injection in language modeling.** In language modeling, embeddings are often perturbed or masked to enhance generalization (Devlin, 2018; Miyato et al., 2017; Zhu et al., 2020). Among various perturbation strategies, Gaussian random noise has been commonly employed for this purpose (Hua et al., 2021; Nukrai et al., 2022; Jain et al., 2024). Specifically, given token embeddings $\mathbf{e}\_i \in \mathbb{R}^d$, noise is injected as $\tilde{\mathbf{e}}_i = \mathbf{e}_i + \boldsymbol{\epsilon}_i$, s.t. $\boldsymbol{\epsilon}_i \sim \mathcal{N}(\mathbf{0}, \sigma^2\mathbf{I})$. This can be understood as encouraging invariance to small perturbations for continuous relaxation of embedding-level augmentation. Our method is inspired by applying random perturbations directly to tokens in the embedding space, rather than to the image input, thereby introducing an additional dimension of augmentation.

**Noise injection in diffusion models.** Denoising Diffusion Models (DDM) (Ho et al., 2020; Song et al., 2020; Nichol & Dhariwal, 2021; Rombach et al., 2022; Ramesh et al., 2021; Saharia et al.,

---

[2]It was inspired by diffusion-style noise injection, but without adopting their training framework.

2022) are based on a forward and reverse process that iteratively adds and removes noise from data. The forward diffusion process gradually corrupts data by adding Gaussian noise. Namely, the forward process is a noise injection step, which is typically described by a Markov chain that adds noise at each step $t$. The data at time step $t$, denoted as $x_t$ is obtained from the original data $x_0$ through $x_t = \sqrt{\alpha_t}x_0 + \sqrt{1 - \alpha_t}\epsilon_t$, where $\alpha_t$ is a schedule that determines how much noise is added at each step, $\epsilon_t$ is the Gaussian noise sampled from $\mathcal{N}(0, I)$, and $x_t$ is the noisy data at time step $t$. Without explicitly employing the diffusion models, the noise injection principles in diffusion model training offer insights into how noise can be injected more effectively.

**Motivation.** Grounded in both prior literature and the observations in Figure 1, our method is driven by three key insights: (1) noise injection has been largely absent and underexplored in the augmentation space; (2) injecting noise at the token level is likely more effective than at the input level; and (3) such an approach can enhance localization ability. More details for the insights will be presented subsequently.

## 2.2 INTRODUCING OUR `DIFFNOISE`

**Design principle.** The core design of `DiffNoise` is guided by the principle that (1) effective regularization should preserve semantic structure while (2) encouraging the model to learn robust and localized representations. We believe that among many options, the noise injection at the embedding space could satisfy the principle.

**Noise injection in the embedding space.** To realize our design principle, we inject noise into the *embedding space* rather than in the input space. Our insight is that input-level perturbations often distort spatial alignment and low-level patterns, which are especially important for fine-grained or structured tasks. In contrast, perturbations at the embedding level are more closely related to control abstract representations, while preserving the essential semantic structure. This choice is supported by recent findings in self-supervised learning and diffusion-based pre-training (Chen et al., 2024; Ho et al., 2020; Choi et al., 2024a), which showcased that injecting noise at the feature level enhances both generalization and localization.

**Employing alpha-blending noise.** To control the strength of injected noise while maintaining semantic fidelity, we adopt an alpha-blending scheme inspired by diffusion training objectives. Rather than directly adding noise, we blend it with the embedding space through a simple formulation: $\tilde{x} = \sqrt{\alpha_t} \cdot x + \sqrt{1 - \alpha_t} \cdot \epsilon$, where $\epsilon \sim \mathcal{N}(0, I)$ is sampled Gaussian noise and $\alpha_t \in (0, 1)$ controls the noise intensity at training step $t$. This ensures that noise is introduced in a smooth and controlled fashion, avoiding the abrupt corruption typically associated with additive noise. By progressively scaling the input and noise, the model is encouraged to adapt to perturbations without compromising the underlying feature structure.

**Implementation.** Algorithm 1 illustrates how `DiffNoise` integrates into the standard fine-tuning pipelines (Dosovitskiy et al., 2020; Touvron et al., 2021; He et al., 2022; Touvron et al., 2022; Heo et al., 2025). Given an existing training setup using strong augmentations (denoted as $\mathcal{R}_b$), DiffNoise requires only *a single additional line*—injecting noise into the embeddings after patch embedding and positional encoding. This simplicity shows the modularity of our method: it acts as a lightweight, plug-and-play augmentation component that operates in the feature space without altering the model architecture or training procedure.

---

**Algorithm 1** Fine-tuning ViTs with `DiffNoise`

---

1: **function** ADDDIFFNOISE$(x, t)$
2:     $s \leftarrow \sqrt{\bar{\alpha}_t}, \quad n \leftarrow \sqrt{1 - \bar{\alpha}_t}$
3:     $\epsilon \sim \mathcal{N}(0, I)$                    ▷ same tensor shape as $x$
4:     **return** $s \cdot x + n \cdot \epsilon$
5: **end function**
6:
7: $x \leftarrow$ DataLoader(ImageNet, augmentation $= \mathcal{R}_b$)
8: $x \leftarrow$ PatchEmbed$(x)$ + PosEmbed
9: $x \leftarrow$ AddDiffNoise$(x, t)$                    ▷ single added line
10: $x \leftarrow$ Encoder$(x)$

---

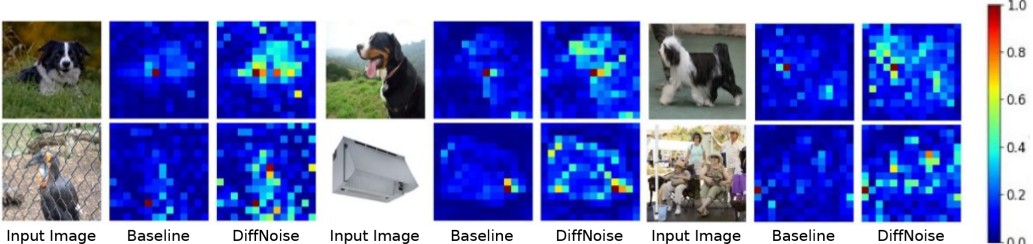

Figure 2: **DiffNoise suppresses high-norm weights for broader attention.** For each input image, the baseline model trained with $\mathcal{R}_b$ (middle) exhibits high-norm tokens (highlighted towards red) focused on semantically irrelevant regions. In contrast, the DiffNoise-augmented model (right) distributes attention more coherently across foreground, reducing isolated peaks and encouraging smoother token interactions.

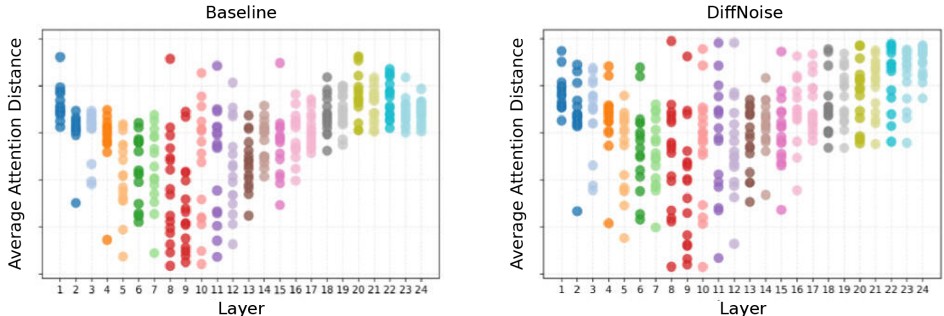

Figure 3: **DiffNoise enjoys broader attention distance.** We measure average attention distance across layers for ViT-L. Baseline model trained with $\mathcal{R}_b$ shows *narrowed* late-layer patterns; DiffNoise acts as an implicit *localizer*, coupling local cues with global context and expanding the attention range into a broader distribution, enhancing feature sharing without architectural changes or extra supervision.

## 3 WHY SIMPLE EMBEDDING-LEVEL NOISE AUGMENTATION WORKS?

This section studies why such a simple noise augmentation could work effectively. We find that DiffNoise (1) acts like a stronger localizer, (2) augments robustness, and (3) serves as a token-aligned regularizer in the embedding space.

### 3.1 DIFFNOISE AS A STRONG LOCALIZER

Here, we study how DiffNoise behaves like an improved model with stronger localization ability by examining attention weights.

**Suppressing high-norms at attentions.** DiffNoise applies spatially localized, diffusion-style perturbations that suppress attention sinks—overly concentrated, high-norm activations on semantically irrelevant regions. As shown in Fig. 2, the baseline model trained with $\mathcal{R}_b$ exhibits sinky patterns with spurious peaks, whereas DiffNoise distributes attention more evenly across task-relevant regions, increasing spatial coverage and reducing false saliency. We attribute this to implicit denoising: localized noise is attenuated as depth increases, dampening noisy activations and enhancing local feature encoding that aligns with object parts.

**Broader attention distance.** To probe representational behavior, we analyze average attention distance across layers for ViT-L trained with and without DiffNoise. As shown in Fig. 3, the baseline model trained with $\mathcal{R}_b$ tends to adopt narrowed attention patterns in later layers, limiting interaction across distant tokens. In contrast, DiffNoise endows the model with a localizer, expands the attention range over diverse layers: it enables attention to couple local cues with global context. The result is a broader, more uniformly distributed attention range across layers—tokens attend more widely over the spatial layout because reliable local features are available to be integrated. This broader receptive behavior promotes stronger feature sharing across tokens and layers, without any architectural changes or explicit supervision.

Table 1: **Improved robustness under distribution shifts.** `DiffNoise` improves out-of-distribution performance across five robustness benchmarks, showing better generalization under domain shift.

| Model | ImageNetV2 | ImageNetSketch | ImageNetA | ObjectNet | ImageNetR |
|-------|-----------|----------------|-----------|-----------|-----------|
| ViT-B | 60.96 | 17.12 | 14.79 | 31.77 | 40.39 |
| + `DiffNoise` | **63.32** | **19.16** | **15.29** | **32.52** | **42.84** |

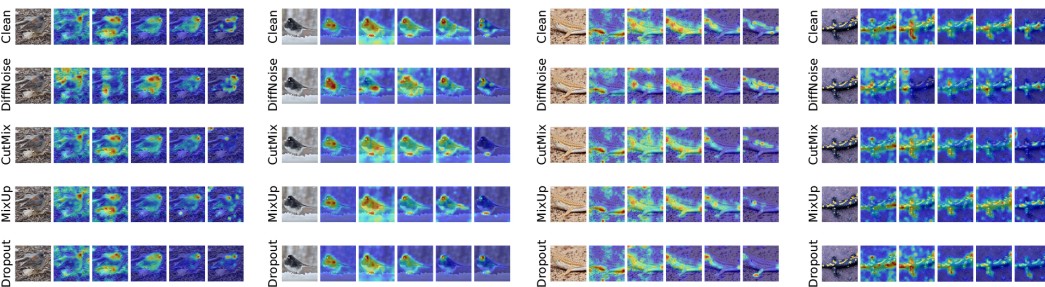

Figure 4: **Robustness test under 20× amplification.** Only `DiffNoise` progressively suppresses structured noise and converges to clean, class-consistent latents; conventional augmentations retain residual artifacts.

**Fine-grained gains from better localization.** On FGVC benchmarks reported in Tab. 4, `DiffNoise` delivers consistent improvements: CUB rises from 79.10 to 80.74, and NABirds from 77.87 to 79.52. These considerable gains align with `DiffNoise` 's localizer effect—suppressing spurious sinks while reinforcing part-level evidence—thereby sharpening boundaries between visually similar subcategories and enhancing fine-grained transfer without additional modules or losses.

## 3.2 DIFFNOISE LERNS MORE ROBUST, GENERALIZED REPRESENTATIONS

We argue that injecting noise at a local neighborhood around each example (at embedding) promotes generalization. This serves as vicinal training in embedding space, smoothing decision boundaries, and discouraging brittle shortcuts. We explore this through experiments on ImageNet distribution shifts and loss surfaces, and further reconstruct noisy images to examine how `DiffNoise` enables the model to learn to handle noise.

**Improved robustness under distribution shifts.** To assess robustness under distribution shifts, we evaluate ViT-B (Dosovitskiy et al., 2020) fine-tuned with and without `DiffNoise` on five challenging ImageNet (Deng et al., 2009) variants: ImageNetV2 (Recht et al., 2019), ImageNetSketch (Wang et al., 2019), ImageNet-A (Hendrycks et al., 2021b), ObjectNet (Barbu et al., 2019), and ImageNet-R (Hendrycks et al., 2021a). As shown in Table 1, `DiffNoise` consistently improves performance across all benchmarks, with gains of +0.75% to +2.45% absolute accuracy. These results indicate that embedding-space noise serves as a complementary regularizer that enhances resilience to input shifts and semantic perturbations, enhancing generalization to unseen domains.

**Generalized representations via flatter minima.** Figure 6 shows the loss landscapes of ViT models trained with and without `DiffNoise`. The baseline model converges to a sharp minimum with steep curvature, indicating sensitivity to perturbations. In contrast, the `DiffNoise`-trained model exhibits a flatter and wider basin, suggesting more stable optimization and better generalization.

**Stress tests with amplified augmentation.** We further conduct two stress tests by amplifying the augmentation strength beyond typical settings. First, in Fig. 4 at 20× strength, only `DiffNoise` recovers clean, class-consistent latents in deeper layers, whereas conventional augmentations leave residual artifacts. Second, directly reconstructing images from feature maps in Fig. 5—under a 50× noise setting—shows that `DiffNoise` maintains sufficient semantic fidelity to approximate the clean input. These findings demonstrate that `DiffNoise` exhibits denoiser-like behavior and sup-

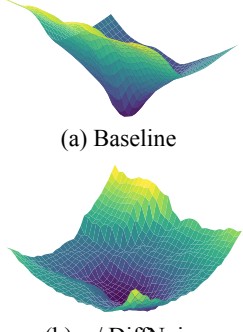

(a) Baseline

(b) w/ DiffNoise

Figure 6: **Robustness via flatter minima.** We plot the loss surfaces: baseline + $\mathcal{R}_b$ (top) and with `DiffNoise` (bottom).

Figure 5: **Reconstructions with `DiffNoise`-trianed models on 50× noise- infected images.** We highlight the striking capability of `DiffNoise`-trained models to process heavily corrupted inputs. Late-layer feature inversions show `DiffNoise` preserves sufficient semantic content to approximate the clean image, which suggests the model has equipped improved noise-robustness.

port the view that `DiffNoise` maintains sufficient semantic fidelity to approximate clean input.

## 3.3 Embedding-space Noise as Regularizer

As a complementary plug-in to standard augmentations, why did prior noise augmentations often fail (see Appendices C and A), yet `DiffNoise` works? We argue that noise placement matters: embedding-space (token-aligned) noise—unlike pixel-space noise that becomes anisotropic after patch embedding—preserves isotropy and alignment, making the regularization effective.

**Why inject at embedding space? Anisotropic vs. isotropic.** The desired noising effect is to inject perturbations that spread uniformly across all embedding dimensions (*i.e.,* token dimensions) as an isotropic perturbation. However, injecting noise at the pixel level yields biased perturbations. Specifically, let $x \in \mathbb{R}^{H \times W \times C}$ and let the patch-embedding operator be (approximately) linear, $\mathcal{P} : \mathbb{R}^{H \times W \times C} \to \mathbb{R}^{N \times d}$ (*e.g.,* a stride-$p$ conv). Injecting pixel-level noise $\epsilon \sim \mathcal{N}(0, \sigma^2 I)$ leads to

$$z_0 = \mathcal{P}(x + \epsilon) = \mathcal{P}x + \underbrace{\mathcal{P}\epsilon}_{\text{embedding noise after patchify}}, \tag{1}$$

so the perturbation at embedding-space is $\epsilon_{\text{tok}} = \mathcal{P}\epsilon$ with covariance

$$\text{Cov}[\epsilon_{\text{tok}}] = \sigma^2 \mathcal{P}\mathcal{P}^\top. \tag{2}$$

Thus, the covariance is no longer isotropic, resulting in biased perturbation that concentrates disproportionately on certain directions or channels rather than being evenly distributed. Moreover, pixel-space corruption disrupts *spatial alignment* before the stem layer (*e.g.,* patchification or set of convolutions), entangling noise across neighboring pixels inside each patch and degrading fine structure that is critical for fine-grained data like FGVC datasets.

In contrast, injecting noise *at embedding space* preserves isotropy, magnitude, and alignment at the level where the model actually computes:

$$z_0 = \mathcal{P}x + \eta_0, \qquad \eta_0 \sim \mathcal{N}(0, \sigma_e^2 I_{Nd}). \tag{3}$$

For a Transformer block with residual pathways:

$$z_{l+1} = z_l + f_l(z_l), \quad \Rightarrow \quad \underbrace{\eta_{l+1}}_{\text{noise}} \approx (I + J_{f_l}(z_l)) \underbrace{\eta_l}_{\text{noise}}, \tag{4}$$

so the skip connection provides a *lossless route* for embedding noise $\eta$ to propagate (*without attenuation by $\mathcal{P}$*). With LayerNorm (LN) centered and scale-learned, the first-order effect of $\eta$ remains observable in the attention logits:

$$A = \text{softmax}\left(\frac{QK^\top}{\sqrt{d}}\right), \quad Q = W_Q \, \text{LN}(z_l), \; K = W_K \, \text{LN}(z_l), \tag{5}$$

thereby producing gradients that explicitly train the network to *denoise* while preserving token boundaries and positional structure intact. Empirically, this yields (i) stronger early-layer diversity (signal present at the level where computation occurs) and (ii) late-layer recovery via residual aggregation—benefits that vanish when noise is injected in pixels and then suppressed by $\mathcal{P}$.

## 4 EXPERIMENT

This section performs comprehensive comparisons between baseline models without augmentation and models trained with `DiffNoise`. Our main analysis covers three major axes: (1) transformer backbone variants (ViT and Swin) including vision-language models (CLIP); (2) self-supervised learning (SSL) frameworks, and (3) extensions to CNNs (ResNet-26/50).

**Ablation studies.** Comprehensive ablations—covering noise injection level, noise type, injection layer, intensity, random vs. fixed schedules, and diffusion-inspired variants—are reported in Appendices C, D, and E.

### 4.1 IMPLEMENTATION DETAILS

**Architecture.** We use Vision Transformers (ViT) (Dosovitskiy et al., 2021) for experiments. The patch size follows the pre-trained architecture, where the ImageNet-21K (Deng et al., 2009) variants typically use a patch size of 16 (*e.g.,*, ViT-B/16). We employ ImageNet-21K (Deng et al., 2009) pre-trained ViTs, SSL-pre-trained ViTs, SwinV2 (Liu et al., 2021), and CLIP (Radford et al., 2021).

**Training setup.** We follow standard fine-tuning protocols from prior work (Dosovitskiy et al., 2020; Touvron et al., 2021; He et al., 2022; Touvron et al., 2022; Heo et al., 2025; Liu et al., 2021). Training is conducted using cosine learning rate scheduling (Loshchilov & Hutter, 2017) with the AdamW optimizer (Loshchilov & Hutter, 2019). Regularization and data augmentation include RandAug (Cubuk et al., 2020), RandErase (DeVries & Taylor, 2017; Zhong et al., 2020), DropPath (Huang et al., 2016), Mixup (Zhang et al., 2017), CutMix (Yun et al., 2019), weight decay, and other standard settings to both baseline and `DiffNoise`-augmented models. Overall, our ImageNet-1K (Deng et al., 2009) training is based on the Timm repository (Wightman, 2019). See Appendix F for detailed experimental settings.

### 4.2 EVALUATION ON TRANSFORMER ARCHITECTURES

We first assess the impact of `DiffNoise` on standard vision transformer models (Dosovitskiy et al., 2020). Experiments are conducted on Vanilla ViT architectures, including ViT-L, ViT-B, and ViT-S, that are pre-trained on ImageNet-1K (Deng et al., 2009) and subsequently fine-tuned with and without `DiffNoise`. Across all ViT architectures, we evaluate `DiffNoise` in comparison to a standard augmentation setup $\mathcal{R}_b$, including CutMix (Yun et al., 2019), MixUp (Zhang et al., 2017), DropPath (Huang et al., 2016), and RandAug (Cubuk et al., 2020), which are commonly used in transformer training pipelines such as timm, as shown in Table 2.

On ViT-B, $\mathcal{R}_b$ lifts Top-1 from 79.02% to 81.17%, while adding other augmentations (Hendrycks et al., 2019; DeVries & Taylor, 2017) yields no further gains due to overlap along the same three axes (Fig. 1). In contrast, `DiffNoise` adds a fourth, previously unexplored axis and raises accuracy to 82.43%, expanding the effective regularization space.

On ViT-S, we observe a similar trend. The baseline achieves 77.79%, while $\mathcal{R}_b$ lead to 78.85%. Incorporating `DiffNoise` results in 79.42%, a relative improvement of +2.09%. For ViT-L, `DiffNoise` improves performance from the baseline of 82.24% to 85.35% when added to the standard augmentation setup. These results indicate that `DiffNoise` complements standard augmentations. Unlike existing methods, which fall into three primary categories in Fig. 1 (b), DiffNoise operates along a distinct regularization axis.

On Swin V2-L (Liu et al., 2021), pre-trained with SimMIM (Xie et al., 2022) on ImageNet-1K (Deng et al., 2009) and then fine-tuned, $\mathcal{R}_b$ reaches 85.21%; adding `DiffNoise` nudges it to 85.30% (+1.45%), consistent with Swin's stronger built-in locality yet confirming `DiffNoise` as a complementary regularizer even for locality-aware backbones. On CLIP (Radford et al., 2021) ViT-B, adding `DiffNoise` lifts accuracy to 84.37% (+1.59% over baseline), indicating that `DiffNoise` transfers beyond vanilla ViTs and remains effective for multi-modal encoders trained with alignment objectives.

**Potential beyond Transformers: extension to CNNs.** We also apply `DiffNoise` to convolutional backbones (ResNet-50/26) on ImageNet and observe consistent gains in Tab. B.1, indicating that the regularization effect is not exclusive to ViTs. See more in Appendix B.

Table 2: **ImageNet-1K (Deng et al., 2009)'s top-1 accuracy and relative improvement.** DiffNoise consistently outperforms standard augmentations by introducing orthogonal regularization in the embedding space. We denote $\mathcal{R}_b$ as the abbreviated recipe: $\mathcal{R}_b$ = CutMix + MixUp + DropPath + RandAug.

| Model | Augmentation/regularization Setup | Top-1 Acc (%) | Relative Gain (%p) |
|---|---|---|---|
| **ViT-B** | + $\mathcal{R}_b$ (CutMix + MixUp + DropPath + RandAug) | 81.17 | +2.71 |
| | + $\mathcal{R}_b$ + AugMix | 81.16 | +2.70 |
| | + $\mathcal{R}_b$ + RandErase | 81.14 | +2.69 |
| | + $\mathcal{R}_b$ + DiffNoise | **82.25** | **+4.08** |
| | + $\mathcal{R}_b$ + AugMix + RandErase | 81.16 | +2.71 |
| | + $\mathcal{R}_b$ + AugMix + DiffNoise | **82.43** | **+4.31** |
| | + $\mathcal{R}_b$ + RandErase + DiffNoise | **82.38** | **+4.25** |
| | + $\mathcal{R}_b$ + AugMix + RandErase + DiffNoise | **82.51** | **+4.41** |
| **ViT-S** | Baseline | 77.79 | – |
| | + $\mathcal{R}_b$ | 78.85 | +1.36 |
| | + $\mathcal{R}_b$ + DiffNoise | **79.42** | **+2.09** |
| **ViT-L** | Baseline | 82.24 | – |
| | + $\mathcal{R}_b$ | 84.71 | +3.00 |
| | + $\mathcal{R}_b$ + DiffNoise | **85.35** | **+3.78** |
| **SwinV2** | Baseline | 84.08 | – |
| | + $\mathcal{R}_b$ | 85.21 | +1.34 |
| | + $\mathcal{R}_b$ + DiffNoise | **85.30** | **+1.45** |
| **CLIP** | Baseline | 83.05 | – |
| | + $\mathcal{R}_b$ | 84.10 | +1.26 |
| | + $\mathcal{R}_b$ + DiffNoise | **84.37** | **+1.59** |
| **ViT-B** | Baseline | 79.02 | – |
| | + CutMix (Yun et al., 2019) | 80.08 | +1.34 |
| | + MixUp (Zhang et al., 2017) | 79.93 | +1.15 |
| | + DropPath (Huang et al., 2016) | 79.65 | +0.80 |
| | + RandAug (Cubuk et al., 2020) | 79.64 | +0.79 |
| | + AutoAug (Cubuk et al., 2019) | 79.63 | +0.77 |
| | + AugMix (Hendrycks et al., 2019) | 79.84 | +1.04 |
| | + RandErase (DeVries & Taylor, 2017) | 79.83 | +1.02 |
| | + DiffNoise | **80.14** | **+1.41** |

Table 3: **Top-1 accuracy on ImageNet-1K** of fine-tuning self-supervised pre-trained models, with and without DiffNoise. We leverage recent state-of-the-art pre-trained models.

| SSL Framework | Model | Pre-training Method | Top-1 Acc (%) |
|---|---|---|---|
| **Masked Image Modeling** | ViT-B | MAE (He et al., 2022) + $\mathcal{R}_b$ | 82.92 |
| | | + DiffNoise | **83.17** |
| | ViT-L | MAE (He et al., 2022) + $\mathcal{R}_b$ | 84.42 |
| | | + DiffNoise | **84.61** |
| | ViT-B | SimMIM (Xie et al., 2022) + $\mathcal{R}_b$ | 83.10 |
| | | + DiffNoise | **83.23** |
| **Diffusion Model-based Masked Image Modeling** | ViT-B | DiffMAE (Wei et al., 2023) + $\mathcal{R}_b$ | 82.18 |
| | | + DiffNoise | **82.50** |
| | ViT-B | MaskDiT (Zheng et al., 2023) + $\mathcal{R}_b$ | 82.89 |
| | | + DiffNoise | **83.14** |
| | ViT-B | DiffMIM (Wei et al., 2023) + $\mathcal{R}_b$ | 83.31 |
| | | + DiffNoise | **83.52** |

## 4.3 EVALUATION WITH MODERN SELF-SUPERVISED LEARNING FRAMEWORKS

Self-supervised learning (SSL)-based pre-training (Xie et al., 2022; He et al., 2022) is now central in vision, particularly for training large Vision Transformers effectively. As capacity grows and data efficiency becomes a bottleneck, SSL exploits unlabeled data to learn transferable representations. In practice, a method's ability to integrate with—and improve—SSL pipelines has become a key test of scalability and relevance.

Table 4: **Evaluation on downstream tasks.** We evaluate our fine-tuned models with `DiffNoise` to assess improvements in generalization ability. `DiffNoise` improves performance consistently across fine-grained visual categorization, semantic segmentation, object detection, and instance segmentation.

| Task | CUB (Acc) | NABirds (Acc) | ADE20K (mIoU) | COCO ($AP^{box}$/$AP^{mask}$) |
|---|---|---|---|---|
| Baseline | 79.10 | 77.87 | 43.12 | 46.17 / 40.21 |
| + DiffNoise | **80.74** | **79.52** | **43.56** | **46.44 / 40.58** |

Moving beyond supervised pre-trained models in the previous section, we apply `DiffNoise` at fine-tuning to SSL pre-trained ImageNet-1K checkpoints from MAE (ViT-B/L) (He et al., 2022), SimMIM (Xie et al., 2022), and diffusion-based SSL (DiffMAE, MaskDiT, DiffMIM) (Wei et al., 2023; Zheng et al., 2023; Choi et al., 2024b) to test our method's generalization to SSL.

The results in Table 3 demonstrate that `DiffNoise` generalizes effectively to the modern SSL frameworks. Across a variety of SSL-pre-trained models, incorporating `DiffNoise` during fine-tuning consistently improves performance over established augmentation baselines, even when standard augmentations are already applied, indicating that `DiffNoise` provides a complementary form of regularization beyond existing methods.

This effect is pronounced in diffusion-based SSL, which has gained increasing prominence recently. For instance, `DiffNoise` lifts DiffMAE (Wei et al., 2023) from 82.18% to 82.50% and DiffMIM (Choi et al., 2024b) from 83.31% to 83.52%. These improvements, ranging from +0.25% to +0.34%, are particularly notable given that they build on already strong SSL baselines. We attribute these gains to reduced pretrain–finetune mismatch: `DiffNoise` injects embedding-space noise and implicitly encourages denoising, aligning fine-tuning with diffusion-style pre-training. Given the ongoing shift toward diffusion-based pre-training in large vision models, `DiffNoise` offers a timely augmentation strategy that integrates with modern SSL methods and yields consistent complementary gains.

### 4.4 EVALUATION ON DOWNSTREAM TASKS

We evaluate DiffNoise on a diverse set of downstream tasks, including fine-grained visual classification (FGVC): CUB (Wah et al., 2011), NABirds (Van Horn et al., 2015)), semantic segmentation (ADE20K (Zhou et al., 2017)), and object detection and instance segmentation (COCO (Lin et al., 2014), as shown in Table 4. DiffNoise improves performance across all tasks, demonstrating its generality beyond image classification.

Gains are strongest on FGVC datasets: +1.64% on CUB (Wah et al., 2011) and +1.65% on NABirds (Van Horn et al., 2015), where subtle part cues (*e.g.,* beak, feather texture) matter and `DiffNoise`'s localization is especially helpful. In semantic segmentation (ADE20K (Zhou et al., 2017)), where spatially coherent semantic understanding is crucial, mIoU rises from 43.12 to 43.56; in object detection and instance segmentation (COCO (Lin et al., 2014)), $AP^{box}$ rises from 46.17 to 46.44 and $AP^{mask}$ rises from 40.21 to 40.58. These improvements suggest that embedding-level perturbation strengthens discriminative features while preserving spatial structure, yielding consistent gains across tasks.

## 5 CONCLUSION

We have presented `DiffNoise`, revisiting noise augmentation particularly for input embeddings, which complements the long-standing methods like Mixup, CutMix, RandAug, and Droppath, despite prior challenges in effectively combining such augmentations with a novel augmentation. Without modifying architectures or introducing additional objectives, it turned out that `DiffNoise` naturally induces denoising through forward propagation, with more precise localization emerging as a beneficial byproduct. Our analysis has further revealed that it flattens the loss landscape and mitigates attention sinks. Extensive experiments show consistent improvements in performance. We believe `DiffNoise` offers a new, orthogonal axis to traditional augmentation strategies, enriching long-standing fixed training recipes with minimal overhead.

**Reproducibility statement.** We conducted all experiments on ImageNet with extra publicly released methods that were all reproducible. Code is available in the Supplementary Material to ensure reproducibility.

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

# Appendix

## A  COMPARISON WITH PRIOR NOISE AUGMENTATIONS

Given the claim that noise-based augmentation enhances locality and thereby representation quality, a natural question is whether prior noise augmentations in fact translate into improved recognition performance. To evaluate this, Tab. A.1 compares several noise families under a matched-strength protocol.

Table A.1: **Comparison with prior noise augmentations.** Pixel-space additive variants yield modest or negative effects; pixel-level alpha blend helps but remains below `DiffNoise`. Placing noise in embedding space (`DiffNoise`) preserves token alignment and delivers the strongest gains.

| Noise Type | Gaussian | Uniform | Speckle | Additive | $\alpha$-blend (pixel-level) | DiffNoise |
|---|---|---|---|---|---|---|
| Acc | 77.30 | 78.84 | 76.53 | 76.38 | 80.17 | **82.25** |

The result shows that prior noise augmentations (Gaussian, Uniform, Speckle, Additive) provide limited gains, while pixel–level $\alpha$-blend is stronger but still lags behind `DiffNoise`. We attribute this gap to *where* noise is applied. Pixel–space perturbations pass through patch embedding and become anisotropic and misaligned at the token level, diversifying early features but impairing late semantic recovery. $\alpha$-blending stabilizes the signal and thus helps, yet it still inherits pre-tokenization alignment loss. By contrast, `DiffNoise` injects noise after tokenization, yielding token-aligned, isotropic perturbations at the level where computation occurs. This preserves spatial structure, explaining the highest Top-1 (82.25).

## B  EXTENSION TO CNNS: EVALUATION ON RESNET

While `DiffNoise` is primarily evaluated on transformer-based architectures, we also assess its applicability to convolutional networks by applying it to ResNet-50 (He et al., 2016) and ResNet-26 (He et al., 2016) on ImageNet (Deng et al., 2009) classification. As shown in Table B.1, `DiffNoise` improves the top-1 accuracy from 79.86% to 80.04% and 73.20% to 73.33%, when added to a standard ResNet-50 and ResNet-26 baseline. This confirms that the regularization effect of `DiffNoise` is not exclusive to transformer models and can extend to CNNs.

Table B.1: **Top-1 accuracy of ResNet (He et al., 2016) on ImageNet classification, with and without `DiffNoise`.** DiffNoise improves performance even on CNN architectures, demonstrating generality beyond transformers.

| Model | Augmentation/regularization Setup | Top-1 Acc (%) |
|---|---|---|
| ResNet-50 (He et al., 2016) | Baseline | 79.86 |
| ResNet-50 (He et al., 2016) | + DiffNoise | **80.04** |
| ResNet-26 (He et al., 2016) | Baseline | 73.20 |
| ResNet-26 (He et al., 2016) | + DiffNoise | **73.33** |

We interpret this improvement as further evidence of `DiffNoise` acting as a general-purpose augmentation method. Unlike traditional augmentations tailored to input-level or region-level transformations, DiffNoise perturbs intermediate features in the embedding space, which also benefits CNN representations by promoting robustness in hidden activations. Nonetheless, the performance gain observed in CNNs is relatively modest compared to the consistent improvements seen in transformers, where attention-based models appear to benefit more from embedding-level regularization. Thus, while `DiffNoise` is broadly applicable, it is particularly effective in models lacking strong inductive biases—such as ViTs (Dosovitskiy et al., 2020)—where denoising behavior and semantic localization play a more critical role.

## C  ABLATION STUDIES

We conduct ablation studies to understand the design choices of `DiffNoise`, summarized in Table C.1.

Table C.1: **Ablation studies with `DiffNoise`.** (a) noise injection level, (b) noise type, (c) noise injection layer, and (d) noise intensity. All reported numbers are ImageNet-1K (Deng et al., 2009) Top-1 Accuracy (%).

| (a) Noise injection level | | (b) Noise type | | (c) Noise injection layer | | (d) Noise intensity | |
|---|---|---|---|---|---|---|---|
| Level | Acc | Type | Acc | Layer | Acc | Intensity ($t$) | Acc |
| Baseline | 79.02 | Baseline | 79.02 | Baseline | 79.02 | Baseline | 79.02 |
| Pixel-level | 80.17 | Additive | 76.38 | Layer 0 | **82.25** | $t = 3$ | 80.46 |
| Embed-level | **82.25** | $\alpha$-blending | **82.25** | Layer 2 | 82.21 | $t = 5$ | 82.16 |
| | | | | Layer 4 | 81.84 | $t = 10$ | **82.25** |
| | | | | Layer 6 | 79.43 | $t = 15$ | 82.19 |
| | | | | | | $t = 20$ | 81.94 |

**Noise injection level.** When noise is injected at the pixel level, performance gains are marginal (80.17%), consistent with prior findings in denoising diffusion models that pixel-level noise corrupts fine-grained structure and is harder to recover. In contrast, embedding-space corruption achieves significantly better results (82.25%).

**Noise type.** We also observe that using additive noise degrades performance (76.38%), even falling below the baseline. This suggests that abrupt perturbations destroy semantic content, whereas alpha-blending, as adopted in `DiffNoise`, integrates noise more smoothly.

**Noise injection layer.** When varying the injection layer, we find that applying noise closer to the input (layer 0 or 2) yields better performance than deeper layers, indicating that early-stage regularization is more beneficial.

**Noise intensity.** Regarding noise intensity, we observe that `DiffNoise` is robust to a wide range of values, with stable performance as long as the noise level is not extremely low or high. Together, these findings validate the two key design choices of `DiffNoise`—blending-based noise injection and embedding-space corruption—as crucial for achieving effective and semantically aligned regularization.

## D  ABLATION STUDY ON RANDOM VS. FIXED NOISE INTENSITY

Inspired by the noise scheduling in diffusion models (Wei et al., 2023; Zheng et al., 2023; Choi et al., 2024b) where noise is injected at random timesteps, we investigate whether randomly varying the noise intensity across training steps can improve model performance. In this setting, we compare fixed noise injection at specific timesteps $t \in 5, 10, 15$ with random sampling of $t$ from various intervals.

As shown in Figure D.1, the best performance is obtained when the noise intensity is fixed at $t = 10$, achieving 82.25% top-1 accuracy. In contrast, randomly sampling $t$ from wider ranges such as [0,15] or [0,10] leads to reduced accuracy. Even narrower random intervals (e.g., [10,15]) slightly underperform compared to the fixed setting.

These results suggest that unlike generative diffusion models (Wei et al., 2023; Zheng et al., 2023; Choi et al., 2024b) where random timestep sampling promotes diverse training signals, random noise intensity in discriminative tasks such as classification may introduce excessive variability. This variability appears to interfere with the model's ability to form localized and stable representations. Fixing the noise intensity at an optimal level encourages more consistent denoising behavior and better spatial focus.

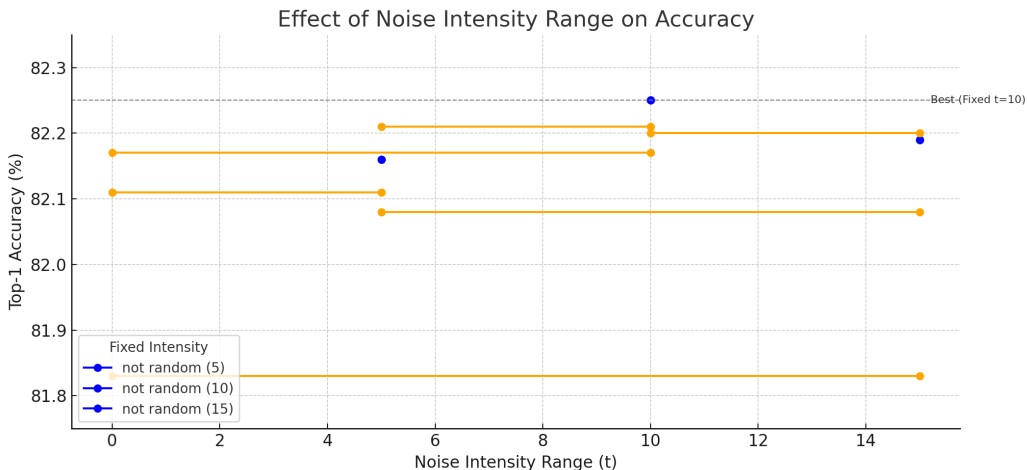

Figure D.1: **Effect of fixed vs. random noise intensity.** Injecting noise with a fixed intensity (particularly at $t = 10$) outperforms random schedules. Unlike generative diffusion models (Wei et al., 2023; Zheng et al., 2023; Choi et al., 2024b) where timestep sampling improves diversity, random noise intensities here degrade localization and lead to less stable representations.

# E ABLATION STUDY ON DIFFUSION-INSPIRED VARIANTS

To further explore how diffusion principles might inform augmentation design, we evaluate several extensions of `DiffNoise` that incorporate elements common in diffusion-based training objectives. The results are summarized in Table E.1.

We first experiment with adding an explicit denoising objective, where the model is trained to reconstruct the clean (unperturbed) embeddings from their noisy counterparts. This setting mimics the forward-reverse formulation in denoising diffusion models. Surprisingly, this explicit supervision yields no performance gain over the original formulation, which relies entirely on implicit denoising. This suggests that allowing the model to learn denoising behavior organically, without human-specified supervision, leads to more effective regularization, revealing an unexpected advantage of self-emergent behavior in deep networks.

Next, we investigate a time-embedding prediction variant. Inspired by the use of timestep conditioning in generative diffusion models, we randomly vary the noise intensity and task the model with predicting the noise level via an additional time-embedder head. This auxiliary prediction head is trained alongside the main task using a scalar regression objective. However, this design consistently underperforms, indicating that exposing the model to variable noise levels and forcing it to regress explicit noise intensities can interfere with the main task learning.

Taken together, these results support the importance of simpler design: the original version of `DiffNoise`, based solely on alpha-blended noise injection into the embedding space, remains the most effective. It encourages implicit denoising without requiring auxiliary heads or explicit supervision.

Table E.1: **Ablation on diffusion-inspired variants.** We evaluate additional designs motivated by a diffusion model, including explicit de-noising and time-embedding prediction. The original formulation, $\alpha$-blending noise with embedding-level injection, achieves the best performance without any auxiliary objectives.

| Method | Top-1 Acc (%) |
|---|---|
| $\alpha$-blending noise + embedding-level noising | **82.25** |
| $\alpha$-blending noise + embedding-level noising + explicit de-noising | 82.21 |
| $\alpha$-blending noise + embedding-level noising + time-embedding prediction | 81.35 |

## F  FINE-TUNING CONFIGURATIONS ACROSS ARCHITECTURES

We summarize the fine-tuning setups for all evaluated models in Tables E.2, E.3, E.4, and E.5. Each table corresponds to a family of architectures: ViT (Dosovitskiy et al., 2020), CLIP (Radford et al., 2021), self-supervised models (MAE (He et al., 2022), SimMIM (Xie et al., 2022)), diffusion-based SSL methods (DiffMAE (Wei et al., 2023), MaskDiT (Zheng et al., 2023), DiffMIM (Choi et al., 2024b)), and ResNet (He et al., 2016).

All models are fine-tuned on ImageNet-1K (Deng et al., 2009) under standardized training pipelines, with consistent augmentation strategies—including Mixup (Zhang et al., 2017), CutMix (Yun et al., 2019), DropPath (Huang et al., 2016), and RandAug (Cubuk et al., 2020)—to ensure fair comparisons. We vary warm-up epochs and layer decay based on model scale and training method, while keeping optimizer and regularization consistent. These tables offer detailed reference points for the reproducibility and comparability of the results presented in our experiments.

## G  RELATED WORK

**Data Augmentation** has been a cornerstone of deep learning in computer vision, primarily for mitigating overfitting and improving model generalization. Early augmentation strategies included simple geometric and photometric transformations such as flipping, cropping, rotation, and color jittering, which proved highly effective in CNN-based models trained on limited data (Krizhevsky et al., 2017). To go beyond heuristic transformations, more structured approaches were proposed, built upon the milestone Dropout (Srivastava et al., 2014). Mixup (Zhang et al., 2017) introduced linear interpolation between input-label pairs, effectively regularizing the decision boundary. CutMix (Yun et al., 2019) improved upon this by pasting patches from one image onto another, preserving semantic context while introducing strong local perturbations. In parallel, DropPath (Huang et al., 2016) applied randomness to the model architecture itself during training, acting as implicit regularizers.

In recent years, policy-based augmentation methods have emerged. AutoAug (Cubuk et al., 2019) and RandAug (Cubuk et al., 2020) learn or sample augmentation policies from data, yielding state-of-the-art results on recognition benchmarks. AugMix (Hendrycks et al., 2019) proposed a compositional and distributionally-robust augmentation scheme blending multiple augmentations while enforcing consistency in predictions. These methods improved robustness and generalization, especially under distribution shift. Despite these advances, many augmentation strategies operate within overlapping regularization spaces, and combinations of strong augmentations often yield diminishing returns—a saturation effect observed in recent studies (DeVries & Taylor, 2017; Zhang et al., 2022). This motivates the search for augmentation techniques introducing orthogonal learning signals and operate along previously untapped axes of regularization.

**Toward Complementary Augmentation beyond Standard Triads.** Mixup (Zhang et al., 2017), CutMix (Yun et al., 2019), DropPath (Huang et al., 2016), and RandAug (Cubuk et al., 2020) have become the de facto augmentation recipe across modern vision training pipelines. These augmentations have been widely adopted as standard components in high-performance vision model implementations, including community-maintained libraries such as timm (Wightman, 2019) and Hugging Face (Wolf et al., 2020), often forming the default augmentation setup for transformer-based architectures. Empirically, their combination yields strong performance across benchmarks.

However, despite their collective success, no augmentation method has yet been shown to complement these in a synergistic manner. That is, additional augmentations often fail to introduce orthogonal regularization signals, instead overlapping with existing methods and offering limited gains. This observation motivates our development of `DiffNoise`, a diffusion model-inspired augmentation strategy designed to operate along an unexplored axis of regularization. `DiffNoise` fills this gap by providing a noise-driven signal that integrates with existing setups.

## H  THE USE OF LLMS.

LLMs were used only for minor language improvements. They were not involved in the conception of the research, experiments, analysis, interpretation, or drafting.

Table E.2: **Fine-tuning settings for ViT (Dosovitskiy et al., 2020) and CLIP (Radford et al., 2021) models.** ViT-{small, base, large} and CLIP are trained on ImageNet-1K (Deng et al., 2009) using AdamW with cosine decay.

| Setting | ViT-S | ViT-B | ViT-L | CLIP ViT-B |
|---|---|---|---|---|
| Optimizer | AdamW | AdamW | AdamW | AdamW |
| Base Learning Rate | 5e-4 | 5e-4 | 1.25e-3 | 1e-3 |
| Weight Decay | 0.05 | 0.05 | 0.05 | 0.05 |
| Layer Decay | 0.65 | 0.65 | 0.9 | 0.6 |
| Optimizer Momentum | $\beta_1 = 0.9, \beta_2 = 0.999$ | $\beta_1 = 0.9, \beta_2 = 0.999$ | $\beta_1 = 0.9, \beta_2 = 0.999$ | $\beta_1 = 0.9, \beta_2 = 0.999$ |
| Learning Rate Schedule | Cosine Decay | Cosine Decay | Cosine Decay | Cosine Decay |
| Drop Path | 0.1 | 0.1 | 0.1 | 0.0 |
| Batch Size | 2048 | 1024 | 1024 | 2048 |
| Warmup Epoch | 5 | 5 | 20 | 20 |
| Training Epoch | 100 | 50 | 100 | 100 |
| RandAug | RandAug(9, 0.5) | RandAug(9, 0.5) | RandAug(9, 0.5) | RandAug(9, 0.5) |
| Mixup | 0.8 | 0.8 | 0.8 | 0.8 |
| Cutmix | 1.0 | 1.0 | 1.0 | 1.0 |
| Label Smoothing | 0.1 | 0.1 | 0.1 | 0.1 |

Table E.3: **Fine-tuning settings for self-supervised pre-trained models (MAE (He et al., 2022), SimMIM (Xie et al., 2022)).** Models are fine-tuned with consistent augmentation and optimization setups.

| Setting | MAE | MAE | SimMIM |
|---|---|---|---|
| Model Size | base | large | base |
| Optimizer | AdamW | AdamW | AdamW |
| Base Learning Rate | 1.0e-3 | 1.0e-3 | 1.25e-3 |
| Weight Decay | 0.05 | 0.05 | 0.05 |
| Layer Decay | 0.65 | 0.75 | 0.9 |
| Optimizer Momentum | $\beta_1 = 0.9, \beta_2 = 0.999$ | $\beta_1 = 0.9, \beta_2 = 0.999$ | $\beta_1 = 0.9, \beta_2 = 0.999$ |
| Learning Rate Schedule | Cosine Decay | Cosine Decay | Cosine Decay |
| Drop Path | 0.1 | 0.2 | 0.1 |
| Batch Size | 2048 | 1024 | 1024 |
| Warmup Epoch | 5 | 5 | 20 |
| Training Epoch | 100 | 50 | 100 |
| RandAug | RandAug(9, 0.5) | RandAug(9, 0.5) | RandAug(9, 0.5) |
| Mixup | 0.8 | 0.8 | 0.8 |
| Cutmix | 1.0 | 1.0 | 1.0 |
| Label Smoothing | 0.1 | 0.1 | 0.1 |
| Random Erasing | 0.25 | 0.25 | 0.25 |

Table E.4: **Fine-tuning settings for diffusion model-based masked image modeling methods (DiffMAE (Wei et al., 2023), MaskDiT (Zheng et al., 2023), DiffMIM (Choi et al., 2024b)).** All models adopt the same augmentation pipeline and optimizer setup. Learning rates and warm-up epochs are scaled to stabilize fine-tuning for high-capacity pre-training.

| Setting | DiffMAE | MaskDiT | DiffMIM |
|---|---|---|---|
| Model Size | base | base | base |
| Optimizer | AdamW | AdamW | AdamW |
| Base Learning Rate | 5e-4 | 5e-4 | 1.25e-3 |
| Weight Decay | 0.05 | 0.05 | 0.05 |
| Layer Decay | 0.65 | 0.65 | 0.65 |
| Optimizer Momentum | $\beta_1 = 0.9, \beta_2 = 0.999$ | $\beta_1 = 0.9, \beta_2 = 0.999$ | $\beta_1 = 0.9, \beta_2 = 0.999$ |
| Learning Rate Schedule | Cosine Decay | Cosine Decay | Cosine Decay |
| Drop Path | 0.1 | 0.1 | 0.1 |
| Batch Size | 4096 | 4096 | 4096 |
| Warmup Epoch | 5 | 5 | 20 |
| Training Epoch | 100 | 100 | 100 |
| RandAug | RandAug(9, 0.5) | RandAug(9, 0.5) | RandAug(9, 0.5) |
| Mixup | 0.8 | 0.8 | 0.8 |
| Cutmix | 1.0 | 1.0 | 1.0 |
| Label Smoothing | 0.1 | 0.1 | 0.1 |
| Random Erasing | 0.25 | 0.25 | 0.25 |

Table E.5: **Fine-tuning settings for ResNet-26 (He et al., 2016) and ResNet-50 (He et al., 2016).** Both models are trained on ImageNet-1K using SGD with cosine decay and strong regularization. These settings are used to evaluate the applicability of `DiffNoise` to CNN architectures.

| Setting | ResNet-26 | ResNet-50 |
|---|---|---|
| Optimizer | SGD | SGD |
| Base Learning Rate | 1e-5 | 1e-4 / 5e-4 |
| Weight Decay | 0.125 | 0.125 |
| Optimizer Momentum / Betas | $\beta_1{=}0.6, \beta_2{=}0.995$ | $\beta_1{=}0.6, \beta_2{=}0.995$ |
| Learning Rate Schedule | Cosine Decay | Cosine Decay |
| Drop Path | 0.1 | 0.1 |
| Dropout | 0.3 | 0.3 |
| Batch Size | 1024 | 2048 |
| Warmup Epochs | 5 | 5 |
| Training Epochs | 100 | 100 |
| RandAug | RandAug(9, 0.5) | - |
| Mixup | 0.2 | 0.2 |
| Cutmix | 1.0 | 1.0 |
| Label Smoothing | 0.1 | 0.1 |

