# OpenReview forum: "Vision Transformers Secretly Crave Noise"
_ICLR.cc/2026/Conference — ICLR 2026 Conference Withdrawn Submission_

### Official Review · Reviewer_69U4 · 2025-10-15

**Soundness:** 2
**Presentation:** 4
**Contribution:** 2
**Rating:** 2
**Confidence:** 3

**Summary:**

The paper introduces DiffNoise, a diffusion-inspired augmentation that injects Gaussian noise into ViT embeddings to improve robustness and localization. It’s simple, plug-and-play, and yields small but consistent gains across ViTs, Swin, CLIP, and SSL models.

**Strengths:**

– Very easy to implement (one-line change).

– Consistent improvements on various architectures.

– Clear motivation and visualization analysis.

**Weaknesses:**

1. **Unsubstantiated “orthogonality” claim.**
   The claim that DiffNoise acts orthogonally to existing augmentations (CutMix, Mixup, RandAug, DropPath) is visually suggested but never verified. There is no quantitative measure showing that it induces independent gradients, feature statistics, or regularization directions. Can the authors demonstrate that the effects of DiffNoise are statistically or geometrically independent from those of existing augmentations? Math in paper doesn't explain this difference.

2. **Qualitative rather than quantitative analysis.**
   The “stronger localizer” explanation is based on attention visualizations and average attention distance plots. However, there are no quantitative localization metrics such as attention entropy, pointing-game accuracy, or CAM overlap. Can localization improvements be verified numerically rather than by visual inspection?

3. **Lack of theoretical grounding.**
   The isotropy argument for embedding-space noise is descriptive and not derived from any formal generalization or stability theory. The paper does not connect DiffNoise to any optimization or information-theoretic principle. Can the authors derive an explicit regularization objective, risk bound, or stability analysis to justify why embedding-space noise should generalize better?

4. **Heuristic $\alpha$-schedule without justification.**
   The best-performing noise level (t = 10) is empirically fixed, but the paper provides no rationale or adaptive mechanism for choosing it. It is unclear how sensitive results are to this parameter. Does DiffNoise over-regularize or under-regularize when $\alpha_t$ is mis-specified, and can it be learned dynamically?

5. **Lack of quantitative loss landscape analysis.**
   The paper claims that DiffNoise leads to flatter minima, but provides only qualitative 2D visualizations. There is no curvature or Hessian-based measurement to substantiate this claim. Can the authors measure sharpness or linear-mode connectivity to confirm the flattening effect?

6. **Ambiguous definition of “localization.”**
    The paper uses “localization” to describe attention spreading, robustness, and fine-grained accuracy, without clearly defining which property is being improved. Which measurable notion of localization does DiffNoise explicitly target, and how does it relate to model robustness or generalization?

**Questions:**

I am not a specialist in computer vision, so there may be parts I have misunderstood. I am open to adjusting my evaluation if the authors can clarify these points or address potential misinterpretations in the rebuttal.

---

### Official Review · Reviewer_bCvi · 2025-10-15

**Soundness:** 2
**Presentation:** 3
**Contribution:** 2
**Rating:** 4
**Confidence:** 4

**Summary:**

The authors proposed a new augmentation strategy termed as DiffNoise, that add noise to tokens instead of pixels, and claimed to have orthogonal augmentation effect on ImageNet-1k with ViT/ResNet and downstream tasks.

**Strengths:**

* Idea is simple and easy to implement.
* Writing is clear.
* Lots of quantitative experiments based on ViT.

**Weaknesses:**

* Idea is not completely new. Although [1] may not be well-known but the idea is very similar to the current manuscript.

[1] Yu, Xiaowei, et al. "Noisynn: Exploring the impact of information entropy change in learning systems." arXiv e-prints (2023): arXiv-2309.
* Lots of experimental demonstration instead of clear first-principle interpretation.
* Experiments is relatively small-scale (mostly on ViT-B and ImageNet-1k).

**Questions:**

* How to compare this paper with the paper mentioned in the weakness part.
* Is the performance gain expected to hold for larger-scale setup?
* Why the performance gain on SSL setup much more smaller than pure classification setup?
* * Specifically, if the performance gain is absorbed into SSL, does that mean we can think about improving SSL methods instead of leveraging this augmentation.

---

### Official Review · Reviewer_hguJ · 2025-10-31

**Soundness:** 2
**Presentation:** 2
**Contribution:** 2
**Rating:** 2
**Confidence:** 4

**Summary:**

This paper introduces DiffNoise, a data augmentation method that injects noise into the embedding space rather than the raw input space, as done in common augmentations such as Mixup, CutMix, and RandAug. The goal of DiffNoise is to address the performance saturation observed when stacking traditional augmentations. The authors hypothesize that existing augmentations may overlap in their regularization effects, leading to diminishing returns. In contrast, injecting noise in the embedding space perturbs abstract representations instead of spatial features, thus preserving the semantic structure of the data. The noise is applied via alpha-blending, similar to diffusion models, which enables smooth control over the intensity of the injected noise.

**Strengths:**

* The proposed method is simple and easy to integrate.

* Evaluation on diverse cases (architecture and training paradigm), with extensive experimentation

**Weaknesses:**

- The writing style is occasionally colloquial, which makes the paper at times difficult to follow (see questions below).

-   Overstatements and modest improvements:
	- (L238–239) — “These considerable gains align with DiffNoise’s localizer effect.” It is difficult to describe the gains as considerable when they are mostly under 2%, even compared to the simple case without augmentation.
	- (Table 1) — “Improved robustness under distribution shifts” only demonstrates that DiffNoise outperforms the simple case without augmentation. This is expected, as perturbations typically enhance robustness. A comparison with baseline augmentation methods is needed to draw valid conclusions about robustness advantages.
	- (Tables 2, 3, 4) — Reporting relative improvement is not a standard way to present performance gains and tends to artificially inflate the results. In absolute terms, the accuracy gains rarely reach 2 %.

-  Inconsistency in terminology and phrasing occasionally obscures meaning. For instance, Figure 2 states “suppresses high-norm weights for broader attention,” while L201–202 says “Suppressing high-norms at attentions.” These seem intended to convey the same idea but are not clearly connected. Moreover, the surrounding paragraph (L201–202) discusses activation norms, which again differs from the title’s phrasing.

**Questions:**

* What is the attention distance in Figure 3 and Line 208–209?

* What is the scale or range of values shown in Figure 3?

* What is meant by localization ability? Line 209

* Regarding the previous point: the paragraph at L201–202 mentions attenuation of localized noise but also refers to localization ability—aren’t these contradictory?

* L263–264: “Stress tests with amplified augmentation” — Is this a novel effect specific to DiffNoise? How does reconstruction with Rb compare to DiffNoise reconstruction?

* Could you clarify the following (L323)?
	* “(signal present at the level where computation occurs)” — Does computation not occur elsewhere as well?
	* “late-layer recovery via residual aggregation—benefits that vanish when noise is injected in pixels and then suppressed by P.”  What exactly is being recovered, and what benefits are being referred to?

* The stated motivation (2) Line 118 — “injecting noise at the token level is likely more effective than at the input level” — is itself a hypothesis to be tested, not a motivation. A motivation should explain why one expects this to work better. I understand the detailed justification appears later, but the current formulation reads as circular.

* Could you elaborate on Equation (4)? Is J the Jacobian?

---

### Note · Authors · 2025-11-14

I have read and agree with the venue's withdrawal policy on behalf of myself and my co-authors.